# Digital transformation and supply chain efficiency improvement: An empirical study from a-share listed companies in China

**Junbo He**[1], **Min Fan**[2]*, **Yaojun Fan**[3]

**1** Xi'an Innovation College of Yan'an University, Xi 'an, China, **2** School of Economics, Lanzhou University, Lanzhou, China, **3** Chinese International College, Dhurakij Pundit University, Bangkok, Thailand

* 1160529881@qq.com

## Abstract

This article thoroughly examines the influence of digital transformation on the efficiency of corporate supply chains. As global economic integration accelerates and technological innovations deepen, digital transformation has become key to enhancing core corporate competitiveness. This research, utilizing data from A-share listed companies in China between 2007 and 2022, analyzes how companies improve supply chain efficiency through digital transformation. Furthermore, the study establishes a theoretical framework that demonstrates how digital transformation facilitates supply chain efficiency from the perspectives of internal governance and external competition. The research indicates that digital transformation plays a key role in significantly enhancing supply chain efficiency. Furthermore, the results of the mechanism analysis confirmed that digital transformation contributes to enhancing corporate supply chain efficiency by improving the level of corporate governance and the degree of market competition. The study also finds that the effect of digital transformation on supply chain efficiency varies with different corporate backgrounds, indicating its heterogeneous impact. Lastly, an analysis of economic consequences shows that the increased supply chain efficiency resulting from digital transformation can reduce future external transaction costs, strengthening the company's market position and financial performance. This research provides strategic guidance for firms to develop robust strategies amid the digital wave and offers strong policy recommendations for promoting digital supply chain management and enhancing market adaptability.

## Introduction

As the global economy becomes more integrated and the competitive landscape intensifies, supply chain management, a critical component of corporate competitive advantage, is increasingly emphasized. The degree of supply chain efficiency plays a crucial role in modern business operations, acting as a vital indicator of a company's overall strength [1]. An efficient supply chain can significantly lower operating costs, including material procurement, inventory, logistics, and time-related opportunity costs [2]. When companies can accurately forecast market demand and adjust their production and inventory accordingly, they minimize waste

**Data Availability Statement:** The sample data used in this study are derived from the financial data of Chinese A-share listed companies between

2007 and 2022, covering critical phases of digital transformation as well as periods of rapid economic development and structural adjustment in China. The data on digital transformation were obtained through an in-depth analysis of information released by the Juchao Information website, which includes various indicators of corporate digitalization efforts. Additionally, we accessed other relevant financial data from the CSMAR database. Given that these data originate from third-party sources, namely Juchao Information website and CSMAR database, we are unable to directly provide these datasets. Researchers interested in this data should apply for access directly from these third-party institutions. Third-party data access instructions are as follows: Digital Transformation Data: Obtained through the Juchao Information website, for specific acquisition methods please refer to the Juchao Information website's relevant page (http://www.cninfo.com.cn/new/index). CSMAR Database Financial Data: Obtained through the CSMAR database, for specific access conditions and methods please refer to the CSMAR official website (https://data.csmar.com/). Since the data are from third-party sources, our research team cannot provide direct download links or datasets. We confirm that, during the course of this study, no data that could not be equally obtained by other researchers or any special access permissions were used.

**Funding:** The author(s) received no specific funding for this work.

**Competing interests:** The authors have declared that no competing interests exist.

from excess inventory and lost sales opportunities due to stockouts. Additionally, supply chain efficiency directly affects a company's responsiveness to market changes [3, 4]. In volatile demand and competitive markets, companies that respond quickly to customer needs gain a competitive edge. Each step of the supply chain, from order processing to product delivery, must be precisely coordinated to fulfill customer demands efficiently and effectively. This agility not only pertains to order response times but also to new product development and market launch timings, allowing companies to seize market opportunities. Traditional supply chains, however, face challenges such as information silos, inaccurate demand forecasting, inefficient inventory management, and lack of transparency [5, 6], which hinder supply chain efficiency and corporate agility, impeding rapid market adaptation and customer satisfaction. Therefore, dismantling information silos, enhancing demand forecasting accuracy, optimizing inventory management, and increasing supply chain transparency and collaboration have become crucial tasks for contemporary corporate supply chain reform.

In today's fast-paced globalization and technological innovation, digital transformation stands as a key driver of economic development [7]. The rapid advancement of information technologies, especially the broad application of the internet, big data, cloud computing, artificial intelligence, and the Internet of Things, is significantly altering operational models across industries, including supply chain management. Digitalization is not just a technological shift but a comprehensive era of integration and optimization of internal and external business resources and processes [8]. Indeed, under this backdrop, digital transformation is a strategic approach to increasing supply chain efficiency. Digital technologies such as IoT, cloud computing, big data analytics, and AI are revolutionizing supply chain management [9]. These technologies enhance data visibility and transparency, improve forecasting accuracy, optimize inventory management, and facilitate supply and demand coordination, thereby increasing overall supply chain efficiency [10]. Thus, digital transformation can potentially change the supply chain governance structure and adjust enterprise supply chain configurations through a process of "digitization—information sharing—resource optimization—organizational change" [11], enhancing supply chain efficiency. However, while the potential of digital transformation is widely recognized, the specific mechanisms, conditions, and economic consequences of its impact on supply chain efficiency need to be further explored.

As a disruptive innovation characterized by technological and organizational change, digital transformation reshapes the fundamental logic of enterprise value creation. The existing literature has delved into the effects of digital transformation on enterprises, such as its impact on total factor productivity [12, 13], corporate performance [14], organizational structure [15, 16], and innovation [17, 18]. Yet, few studies have explored the economic outcomes of digital transformation from the standpoint of supply chain efficiency [19–22]. While existing studies have provided invaluable insights, there are evident gaps in several key areas that remain unaddressed. Initially, there is a general lack of in-depth analysis on the intrinsic mechanisms through which digital transformation enhances supply chain efficiency, particularly on how efficiency can be significantly improved through the optimization of internal governance and the intensification of external market competition. Furthermore, the analysis of the heterogeneous effects of digital transformation on supply chain efficiency is insufficient, lacking a detailed exploration of the variations in effects across different corporate backgrounds, such as company size, environmental performance, and the quality of information disclosure. Lastly, few studies have evaluated the economic consequences of digital transformation in terms of its impact on external transaction costs and market positioning through the enhancement of supply chain efficiency. To bridge these gaps, our research conducted a detailed analysis based on data from Chinese A-share listed companies from 2007 to 2022. Employing quantitative research methods, including regression analysis and other statistical techniques, this study

assessed the specific impact of digital transformation on supply chain efficiency. Through mechanism and heterogeneity analysis, it revealed the pathways of impact and the variance in effects under different corporate backgrounds. Our robustness tests confirmed the reliability of our findings, indicating a significant positive impact of digital transformation on corporate supply chain efficiency, especially notable in companies with low environmental performance, small scale, and poor information disclosure quality. Moreover, the analysis of economic consequences showed that digital transformation, by enhancing supply chain efficiency, could reduce future external transaction costs, thereby strengthening the market position and financial performance of the firm.

## Literature review and theoretical analysis

### Literature review

Supply chain efficiency refers to the extent to which the least amount of resources and inputs are used to meet customer needs during supply chain management. It involves optimizing various aspects of supply chain operations, including procurement, production, storage, distribution, and information flow, to realize the maximum value output of products and services at the lowest possible cost and time. Existing literature posits that within the context of supply chain efficiency, the focus is on how to reduce waste, increase operational speed, lower costs, and simultaneously maintain or improve service quality [23–25]. This often involves continuous improvement of supply chain processes [26, 27], including the adoption of advanced technology and management methods [28], as well as close cooperation with suppliers and distributors to ensure the smooth operation of the entire supply chain. Additionally, literature suggests that improving supply chain efficiency also depends on keen insights into market dynamics and consumer demands [29]. Businesses need to constantly adjust their supply chain strategies to adapt to rapidly changing market conditions and consumer preferences. For instance, by implementing real-time data analysis and monitoring market trends, businesses can respond more quickly to market changes, thus increasing the adaptability and flexibility of the supply chain [30].

As for the economic consequences of supply chain efficiency, the academic community has conducted in-depth studies. Research by Didenot & Díaz (2012) [31] points out that integration and collaboration within the supply chain can bring significant benefits to companies, including added value, efficiency creation, and customer satisfaction. These benefits are reflected in reduced inventory, improved service delivery and quality, and shortened product development cycles. The improvement of supply chain efficiency may also lead to an increase in production scale, helping enterprises achieve economies of scale and further reduce unit costs [2]. Moreover, enhancing supply chain efficiency often accompanies a more rational use of resources, contributing to enterprises' goals of environmental sustainability, which is of increasing concern to consumers and regulatory bodies [32]. On a broader level, improvements in supply chain efficiency can enhance a nation's economic competitiveness, promote foreign trade, create job opportunities, and potentially bring greater tax revenues [33, 34]. In summary, improving supply chain efficiency is a key factor for sustainable growth of businesses, not only improving the financial performance of enterprises but also generating positive impacts in the broader environmental and economic context.

With the deepening development of cutting-edge technologies such as artificial intelligence, blockchain, big data, and cloud computing, they have transformed from concepts into practical tools that drive enterprise transformation and innovation [35]. These technologies are increasingly becoming the core of corporate strategic planning because they greatly enhance efficiency, reduce costs, and bring competitive advantages. In production, artificial intelligence

and machine learning are used for predictive maintenance, optimizing supply chains, and even adjusting parameters in real-time during the production process to ensure quality and efficiency [36, 37]. On the decision-making level, big data analytics offers managers unprecedented insights, enabling them to make more informed and precise strategic decisions based on vast amounts of data [38]. In operational interaction, cloud computing provides a seamless platform for collaboration between different regions and departments, making the flow of information faster and more secure, greatly improving work efficiency and response times [39]. Blockchain technology plays a key role in enhancing the transparency and security of transactions and in tracking the complete journey of products from start to finish [40]. Therefore, existing literature recognizes that the integration of these technologies has not just changed individual aspects but has led to comprehensive digital transformation throughout the organization, creating new value chains for enterprises. While cutting-edge technologies such as artificial intelligence, blockchain, big data, and cloud computing are widely recognized as key drivers for corporate transformation and innovation, the discussion on how these technologies can be specifically applied to maximize efficiency in accordance with a company's unique needs and supply chain structure remains somewhat limited in existing literature. Particularly, the integration of such technologies with current supply chain management practices and the overcoming of challenges and resistance during implementation are critical directions for future research.

So, can enterprise digital transformation drive supply chain efficiency? Although preliminary investigations into the relationship between corporate digital transformation and the enhancement of supply chain efficiency have been conducted, there is a relative lack of empirical examination into the intrinsic mechanisms, especially on how digital transformation can enhance supply chain efficiency by altering internal management processes, improving decision-making efficiency, and optimizing external partnerships. Therefore, future studies need to enrich and systematically expand in terms of research perspectives, mechanisms, and scopes, particularly in exploring the differential impacts of digital transformation across various industries and company sizes, as well as how to effectively integrate emerging technologies with traditional supply chain management practices.

## Analysis of the impact of digital transformation on corporate supply chain efficiency

Digitalization is increasingly becoming a key factor in shaping modern supply chains, with the application of digital technologies extending beyond traditional fields of automation and information technology to every link of the supply chain, ushering in a new era of digitalization, networking, and intelligence for supply chain management. For example, through real-time data analysis, companies can rapidly respond to market changes and adjust their supply chain strategies in time to cope with sudden events or market fluctuations [41]. On one hand, digital technologies such as the Internet of Things (IoT) and cloud computing have made information flow between different parts of the supply chain more smooth, achieving efficient integration between upstream and downstream enterprises [42, 43]. This connectivity not only increases the transparency of information but also creates conditions for closer cooperation and coordination. On the other hand, big data analysis and forecasting models enable companies to more accurately predict market demand, thereby optimizing inventory levels and production plans to achieve precise matching between supply and demand [44]. Additionally, the application of technologies like blockchain increases trust among supply chain parties by providing a shared, immutable data platform, where all participants can access real-time, consistent information, supporting more efficient collaborative work [45]. Overall, digital transformation indicates that supply chain management is becoming more refined and dynamic, where enterprises can

use these technologies to achieve optimal resource allocation, increase operational flexibility, reduce waste, and ultimately significantly improve the efficiency and benefits of the entire supply chain. Therefore, the following hypothesis H1 is proposed.

H1: As enterprises delve deeper into digital transformation, their supply chain efficiency is expected to significantly improve. It can be anticipated that by adopting digital technologies such as the Internet of Things (IoT), big data analytics, and cloud computing, companies will be able to optimize various segments of their supply chain. This optimization includes, but is not limited to, enhancing the transparency of the supply chain, increasing the efficiency of logistics and inventory management, and strengthening collaboration among supply chain partners. These measures are expected to directly reduce the operational costs of the supply chain and shorten product delivery times, thereby overall enhancing supply chain efficiency.

## Digital transformation, governance level, and corporate supply chain efficiency

From an internal corporate perspective, enhancing the level of corporate governance is crucial for improving supply chain efficiency [46]. Good corporate governance provides a structured framework and processes for supply chain management, ensuring transparency and compliance of supply chain activities, and promoting efficient resource use and risk management. Firstly, the enhancement of corporate governance usually comes with an emphasis on compliance [47], ensuring that supply chain operations adhere to laws, regulations, and industry standards, reducing the risk and cost of non-compliance. Secondly, improved governance means more open and transparent decision-making processes, where decision-makers can utilize comprehensive data and analytical tools to make consistent and strategic decisions. Thirdly, a clear governance mechanism helps define the responsibilities of various roles and departments, fostering inter-departmental coordination and cooperation, increasing the flexibility and response speed of the supply chain. Furthermore, a sound governance structure also strengthens internal control mechanisms [48], ensuring that supply chain processes are properly supervised and managed, thus improving overall supply chain efficiency.

Driven by digitalization, corporate governance is no longer a static set of rules and regulations but a dynamic, data-driven management system that can quickly adapt to environmental changes, supporting continuous improvement and business innovation. Thus, digital transformation is a key catalyst in advancing modern corporate governance [49, 50], providing necessary tools and capabilities for enterprises to maintain competitiveness in the complex and volatile global economic environment. For example, artificial intelligence can be used to analyze market trends and consumer behavior to guide strategic decisions [51]; blockchain technology can create a secure, immutable record in the supply chain, thereby increasing the transparency and traceability of transactions [52]; and cloud computing provides flexible resource allocation and data storage solutions, facilitating collaboration across departments and regions [53]. Additionally, digital transformation, through automated processes and real-time monitoring systems, helps enterprises identify and respond to risks promptly, enhancing internal control and compliance checking capabilities [54], thereby promoting the improvement of corporate governance level. In summary, the following hypothesis H2 is proposed.

H2: Digital transformation can enhance supply chain efficiency by improving governance levels, with internal governance playing a mediating role between digital transformation and supply chain efficiency. Specifically, digital transformation enhances a company's internal governance structure (for example, by improving the efficiency and transparency of decision-making, better risk management, and strengthening oversight mechanisms), providing a solid

foundation for the optimization of supply chain management. Optimized internal governance further enables companies to more effectively implement digital technologies and strategies, thus enhancing supply chain efficiency.

## Digital transformation, market competition, and supply chain efficiency

From an external corporate perspective, market competition is a key factor that drives the improvement of supply chain efficiency and plays an undeniable role in promoting this. A competitive market environment forces enterprises to constantly seek more efficient supply chain strategies [55] to reduce costs, improve response speed, and enhance customer service quality. Under this pressure, enterprises are propelled to innovate their logistics, inventory management, procurement processes, and distribution strategies to better meet customer needs while maintaining or increasing market share. Market competition prompts enterprises to continuously evaluate and compare their supply chain efficiency with industry best practices, thereby identifying potential areas for efficiency improvement. It also encourages businesses to utilize new technologies and process innovations to increase transparency and flexibility, reducing bottlenecks and redundancies in the supply chain. Furthermore, the competitive environment also promotes the formation of cooperation and partnerships, as companies may seek partners to jointly optimize the supply chain, reduce costs, and improve efficiency through collective expertise [56]. This cooperation can extend to supplier relationship management, promoting the sharing of best practices and even joint investment in supply chain innovation projects.

Digitalization plays a critical role in the contemporary business environment, acting as a significant force in fostering market competition [57]. By introducing advanced information technologies, companies can improve and innovate their business models, enhance operational efficiency, reduce costs, and provide a better customer experience. Digitalization allows businesses to collect and analyze vast amounts of data, thereby better understanding market trends and consumer behavior, achieving precise marketing and personalized services [58]. These improvements enable businesses to quickly adapt to market changes, anticipate and meet consumer needs in advance, thus gaining an advantage in competitive markets. Additionally, digitalization provides a platform for inter-enterprise collaboration, facilitating knowledge sharing and innovation, which significantly accelerates the development and launch of new products, enhancing the innovation capacity of enterprises. Digitalization also enables small and medium-sized enterprises to compete with large corporations by offering specialized services or entering niche markets, thereby increasing market dynamism and diversity. Globally, digitalization expands businesses' market coverage, allowing them to transcend geographical boundaries and reach a wider customer base [59], promoting international trade and enhancing cross-cultural and transnational market competition. Therefore, digital transformation is not only key to driving the competitiveness of individual enterprises but also a driving force for the entire market competition and the development of the global business environment. Hence, the following hypothesis H3 is proposed.

H3: Digital transformation can enhance supply chain efficiency by intensifying market competition, with market competition serving as another mediating variable in the impact of digital transformation on supply chain efficiency. Specifically, a company's digital transformation not only improves its internal operational efficiency but also enhances its competitiveness in the market. This compels competitors to adopt similar digital measures to maintain their competitive position. Such heightened market competition prompts companies to further optimize their supply chain management, aiming for cost reduction and faster response times, thereby overall enhancing supply chain efficiency.

**Table 1. Descriptive statistics.**

| VarName | Obs | Mean | SD | Median | Min | Max |
|---|---|---|---|---|---|---|
| stock | 40321 | -4.4463 | 1.2509 | -4.5179 | -7.6885 | 0.1795 |
| dig | 40321 | 1.2907 | 1.3936 | 1.0986 | 0.0000 | 5.1059 |
| nature | 40321 | 0.1463 | 0.3534 | 0.0000 | 0.0000 | 1.0000 |
| lev | 40321 | 0.4267 | 0.2105 | 0.4178 | 0.0504 | 0.9519 |
| roa | 40321 | 0.0377 | 0.0651 | 0.0385 | -0.2733 | 0.2136 |
| cash | 40321 | 0.1684 | 0.1342 | 0.1292 | 0.0094 | 0.6893 |
| growth | 40321 | 0.3519 | 0.9484 | 0.1218 | -0.6929 | 6.6210 |
| age | 40321 | 2.2194 | 0.7646 | 2.3979 | 0.6931 | 3.3673 |
| tobinq | 40321 | 2.0433 | 1.3245 | 1.6190 | 0.8544 | 8.9278 |
| top1 | 40321 | 0.3462 | 0.1488 | 0.3243 | 0.0850 | 0.7499 |
| dpe | 40321 | 0.3745 | 0.0533 | 0.3333 | 0.3000 | 0.5714 |
| board | 40321 | 2.2440 | 0.1772 | 2.3026 | 1.7918 | 2.7726 |

## Research design

### Data sources

The sample employed in this study encompasses the financial data of Chinese A-share listed companies from 2007 to 2022, a period critical for digital transformation, alongside China's rapid economic development and structural adjustment. Data on digital transformation were acquired through an in-depth analysis of information released by the Juchao Information website, covering various indicators of enterprises' digitalization efforts. To ensure the accuracy and reliability of our research, we sourced additional relevant financial data from the CSMAR database. During the data processing phase, we applied a series of stringent selection criteria: firstly, excluding samples from all ST and *ST companies to avoid the impact of financial distress on the research results; secondly, samples from the financial sector were excluded due to the unique characteristics of the industry that might cause data bias; thirdly, all samples with missing values were eliminated to ensure the completeness of the analysis; finally, all continuous variables were winsorized to mitigate the influence of extreme values. These meticulous data processing steps ensured that the final 40,321 observations obtained were highly representative and credible. The descriptive statistics of the main variables are shown in Table 1.

### Variable selection

**Dependent variable.** Supply Chain Efficiency (stock). Supply chain efficiency measures the fluidity of the flow of products and services within a company's supply chain. The key lies in accelerating the exchange and trade frequency between upstream and downstream enterprises, which is manifested in the smooth circulation and turnover of products and services. Following the study by Zhang et al. (2023) [21], this paper reflects supply chain efficiency based on the number of inventory turnover days. Specifically, it is calculated as ln(365/inventory turnover rate), with a smaller value indicating higher supply chain efficiency. For ease of comparison, this value is multiplied by -1.

**Core explanatory variable.** Digital Transformation (dig). Digital transformation refers to the integration of advanced digital technologies, such as artificial intelligence, big data, cloud computing, and blockchain, into business operations and management to enhance business performance and market competitiveness. According to existing literature, the method of using the frequency or proportion of digital-related keywords in corporate annual reports as a

**Table 2. Variable definitions.**

| Variable | Symbol | Definition |
| --- | --- | --- |
| Digital transformation | stock | ln (365/ Inventory turnover) |
| Digital transformation | dig | ln (Word frequency +1) |
| Property right nature | nature | If it is a state-owned enterprise, the value is 1, otherwise it is 0 |
| Asset-liability ratio | lev | Total liabilities/total assets |
| Return on assets | roa | Net profit/total assets |
| Operating cash flow | cash | Net cash flow/total assets |
| Sales growth rate | growth | Revenue growth of main business in the current year/last year |
| Enterprise age | age | Ratio of main business income |
| Tobin's Q value | tobinq | ln (Age to market +1) |
| Ownership concentration | top1 | Market value/replacement capital |
| Proportion of independent directors | dpe | The proportion of the largest shareholder |
| Board size | board | Number of independent directors/Number of board members |

measure of digitalization has been widely applied. Drawing on the study by Wu et al. (2021) [17], this paper compiles statistics on the frequency of 76 digital-related terms across five dimensions: artificial intelligence technology, big data technology, cloud computing technology, blockchain technology, and digital technology application. The total frequency of terms across these six areas is used as an indicator of the level of digitalization in circulation enterprises, and the natural logarithm is taken after adding 1.

**Control variables.** Based on relevant literature [19–21], this paper selects factors such as the nature of property rights (nature), debt-to-asset ratio (lev), return on assets (roa), operational cash flow (cash), sales growth rate (growth), company age (age), Tobin's Q value (tobinq), ownership concentration (top1), proportion of independent directors (dpe), and board size (board) as control variables in the main regression model. These variables are chosen to eliminate the impact of heterogeneity in corporate characteristics on supply chain efficiency. Definitions of variables are shown in Table 2.

## Model construction

In this study, we referred to the work of Zhang et al. (2023) [21] to construct a model exploring the impact of digital transformation on corporate supply chain efficiency. The selection of this model was based on several key considerations: first, by including firm fixed effects and time fixed effects, we thoroughly considered the potential impact of intrinsic corporate characteristics and time series factors on supply chain efficiency, aiming to enhance the accuracy and reliability of the regression results. Secondly, the carefully selected control variables in the model help to exclude the interference of other potential factors, ensuring an accurate assessment of the impact of digital transformation. Moreover, the standard errors of the regression coefficients were adjusted for clustering at the firm level, taking into account the correlation between different companies, which strengthens the robustness of the model estimates.

$$stock_{i,t} = \alpha_0 + \alpha_1 dig_{i,t} + \delta X + \gamma_i + \omega_t + \varepsilon_{i,t} \tag{1}$$

In Eq (1), stock represents the supply chain efficiency of the enterprise, dig is the level of digitalization, X represents control variables, $\gamma_i$ is the enterprise fixed effect, and $\omega_t$ is the time fixed effect.

**Table 3. Baseline regression.**

|  | (1) | (2) |
|---|---|---|
|  | stock | stock |
| dig | 0.0316*** | 0.0378*** |
|  | (0.0104) | (0.0104) |
| nature |  | 0.0279 |
|  |  | (0.0198) |
| lev |  | -0.0466 |
|  |  | (0.0872) |
| roa |  | 1.0156*** |
|  |  | (0.1156) |
| cash |  | 0.4666*** |
|  |  | (0.0678) |
| growth |  | -0.0210*** |
|  |  | (0.0077) |
| age |  | -0.0439 |
|  |  | (0.0318) |
| tobinq |  | 0.0114 |
|  |  | (0.0081) |
| top1 |  | 0.0260 |
|  |  | (0.1499) |
| dpe |  | -0.1649 |
|  |  | (0.1973) |
| board |  | -0.0125 |
|  |  | (0.0741) |
| _cons | -4.4871*** | -4.4338*** |
|  | (0.0135) | (0.2456) |
| Control | NO | YES |
| Firm_FE | YES | YES |
| Year_FE | YES | YES |
| Obs | 40321 | 40321 |
| r2_a | 0.7837 | 0.7883 |

Note

*, ** and *** passed the significance test at the level of 10%, 5% and 1% respectively, the same below

# Empirical results

## Baseline analysis

Table 3 presents the baseline regression results for the impact of digital transformation on enterprise supply chain efficiency. Column (1) shows the regression results without control variables, where the coefficient for dig is 0.0316 and significant at the 1% level. Column (2) includes control variables, and the coefficient for dig is 0.0378, also significant at the 1% level. It is evident that digital transformation significantly enhances enterprise supply chain efficiency. These results underscore the importance of considering digital transformation in corporate strategy formulation. Improvements in the level of digitalization, by refining inventory management, optimizing material flow, enhancing information sharing, and improving customer service, assist enterprises in rapidly adjusting their supply chain strategies in a dynamically changing market environment, thereby enhancing overall supply chain efficiency. Hypothesis H1 is thus verified.

## Robustness test

To enhance the robustness of the regression results, this study conducts a series of robustness checks. Firstly, the core explanatory variable is recalculated. Drawing on the study by Zhao et al. (2021) [12], this research compiles statistics on the frequency of 99 digital-related terms across four dimensions: digital technology application, internet business models, smart manufacturing, and modern information systems (dig2) as a proxy indicator for the level of digitalization. The regression results are shown in column (1) of Table 4, where the coefficient of dig2 remains positive, indicating that digital transformation can promote the improvement of enterprise supply chain efficiency.

Secondly, considering the possible time lag in the impact of digital transformation on enterprise supply chain efficiency, the study lags the core explanatory variable and control variables by one period before re-running the regression. The results, as shown in column (2) of Table 4, indicate that the coefficient for Ldig remains positive. This demonstrates that even after considering the potential lag effect, digital transformation still enhances enterprise supply chain efficiency.

Thirdly, to reduce the endogeneity problems caused by omitted variables, the study adds two control variables, company size (size) and total factor productivity (tfp), and reruns the regression. Company size is measured by the logarithm of total assets, and total factor productivity is measured using the LP method. The results are shown in column (3) of Table 4 and remain robust.

Next, considering that digital transformation might have been affected by the 2008 global financial crisis, neglecting such significant factors could lead to endogeneity issues [60]. Furthermore, considering the after-effects of the financial crisis, the study excludes data from 2008–2010 and re-tests. The results, as shown in column (4) of Table 4, are also robust.

Lastly, to account for potential reverse causality issues, the study constructs an instrumental variable and uses the two-stage least squares method to re-run the regression. Specifically, the initial share mobility instrumental variable method is adopted, using the product of the initial

**Table 4. Robustness test.**

|          | (1)        | (2)        | (3)        | (4)        | (5)       |
|----------|------------|------------|------------|------------|-----------|
|          | stock      | stock      | stock      | stock      | stock     |
| dig      |            |            | 0.0204**   | 0.0401***  | 1.0602**  |
|          |            |            | (0.0101)   | (0.0096)   | (0.4471)  |
| Ldig     |            | 0.0095**   |            |            |           |
|          |            | (0.0045)   |            |            |           |
| dig2     | 0.0542***  |            |            |            |           |
|          | (0.0116)   |            |            |            |           |
| size     |            |            | -0.2911*** |            |           |
|          |            |            | (0.0328)   |            |           |
| tfp      |            |            | 0.5331***  |            |           |
|          |            |            | (0.0271)   |            |           |
| _cons    | -4.5108*** | -4.4564*** | -2.2537*** | -4.5489*** |           |
|          | (0.2475)   | (0.0005)   | (0.6540)   | (0.2667)   |           |
| Control  | YES        | YES        | YES        | YES        | YES       |
| Firm_FE  | YES        | YES        | YES        | YES        | YES       |
| Year_FE  | YES        | YES        | YES        | YES        | YES       |
| Obs      | 40321      | 34804      | 36502      | 34890      | 27453     |
| r2_a     | 0.7885     | 0.7955     | 0.8103     | 0.8199     | -1.6848   |

share of the analysis unit and the overall growth rate to construct the instrumental variable [60]. On the one hand, the annual growth rate of the average level of digital transformation for all companies is calculated as the overall growth rate. On the other hand, the average level of digital transformation from the previous year in other companies within the same industry and province as each company is calculated as the initial share of the analysis unit. The product of these two figures is used as the instrumental variable for digital transformation. Since the success and quality of a company's digital transformation are closely related to the level of digitalization in its location and industry, this satisfies the relevance condition. Additionally, the construction of the mobility share and the use of the previous year's sample can effectively mitigate the shortcomings of insufficient exogeneity. The second-stage regression results are shown in column (5) of Table 4, where the coefficient for dig remains significantly positive, indicating that digital transformation can still promote the improvement of enterprise supply chain efficiency. The Cragg-Donald Wald F statistic is 28.598, passing the weak instrument variable test.

## Mechanism analysis

To assess the mechanisms through which digitalization influences corporate supply chain efficiency, and given the evident causal inference flaws in the three-stage mediation mechanism test (Jiang, 2022) [61], this study follows the approach of Wu and Yao (2023) [11]. We construct the interaction model depicted in Eq (2) to examine the effects of corporate governance level and market competition degree. If digital transformation promotes supply chain efficiency by enhancing corporate governance levels and market competition, then digital transformation should be more beneficial in promoting supply chain efficiency for companies with higher levels of corporate governance and market competition. Such companies' digital transformation will have a more pronounced effect on improving supply chain efficiency. Therefore, the following interaction term model is constructed to test the causal mechanism by which digital transformation promotes supply chain efficiency by reducing inventory turnover days through enhancing corporate governance levels and market competition degrees. In this model, chain represents the mechanism variable, either corporate governance level or market competition degree. If digital transformation indeed helps to improve corporate governance level or market competition degree, and thereby promote supply chain efficiency, then the coefficient of the interaction term should be significantly positive.

$$stock_{i,t} = \alpha_0 + \alpha_1 dig_{i,t} + \alpha_2 chain + \alpha_3 dig*chain + \delta X + \gamma_i + \omega_t + \varepsilon_{i,t} \tag{2}$$

Firstly, the study tests whether digital transformation enhances enterprise supply chain efficiency by improving corporate governance levels. The level of corporate internal governance reflects the efficiency and effectiveness of a company's management structure, particularly in its capacity to promote transparency, prevent conflicts of interest, and protect the rights of investors. Specifically, following the approach of Zhou et al. (2020) [62], principal component analysis is used to integrate eight indicators—duality of roles, proportion of independent directors, board shareholding ratio, senior management shareholding ratio, the largest shareholder's shareholding ratio, board size, supervisory board size, and senior management remuneration—into a corporate governance level index (govern). A higher value represents a higher level of corporate governance. The regression results, as shown in column (1) of Table 5, indicate that the coefficient for the interaction term of digital transformation and corporate governance level is significantly positive. This suggests that digital transformation can enhance enterprise supply chain efficiency by improving corporate governance levels. When companies adopt digital strategies, their governance structures may become more efficient due

**Table 5. Mechanism test.**

|  | (1) | (2) |
|---|---|---|
|  | stock | stock |
| dig | 0.0401*** | 0.0009 |
|  | (0.0124) | (0.0117) |
| govern | -0.0684** |  |
|  | (0.0333) |  |
| *dig*\*govern | 0.0293*** |  |
|  | (0.0107) |  |
| mc |  | 0.0021 |
|  |  | (0.0985) |
| *dig*\*mc |  | 0.2265*** |
|  |  | (0.0684) |
| _cons | -4.4302*** | -4.4026*** |
|  | (0.2932) | (0.2400) |
| Control | YES | YES |
| Firm_FE | YES | YES |
| Year_FE | YES | YES |
| Obs | 25120 | 40049 |
| r2_a | 0.7923 | 0.7893 |

to better data analysis and process automation, which in turn enables companies to respond more quickly to market changes and manage the supply chain more effectively.

Secondly, the study tests whether digital transformation enhances enterprise supply chain efficiency by increasing market competition levels. The degree of market competition refers to the intensity of competition and the number of competitors that a company faces within its industry. Specifically, following the approach of Wu et al. (2023) [63], the Herfindahl index is used to measure the degree of market competition (mc). The Herfindahl index is a commonly used indicator to assess industry concentration and the intensity of market competition. A lower value indicates a greater number of companies in the market and higher competition; for ease of comparison, this value is multiplied by -1. The regression results, as shown in column (2) of Table 5, indicate that the coefficient for the interaction term of digital transformation and market competition level is significantly positive. This suggests that digital transformation can enhance enterprise supply chain efficiency by increasing market competition levels. As market competition intensifies, companies are more actively adopting digital tools and strategies, such as online platforms, automation technologies, and data analysis tools, to enhance their supply chain operational efficiency, reduce costs, and improve customer service quality, adapting rapidly to changes in market demand.

## Heterogeneity analysis

**The impact of environmental performance differences on outcomes.** To further explore, the study examines whether the effect of digitalization on enhancing enterprise supply chain efficiency varies under different circumstances. First, it assesses whether the results change under different environmental performance scenarios. The total sample is divided into high and low environmental performance groups for regression. Specifically, this study, following the approach of Li et al. (2023) [60], combines qualitative and quantitative measures to evaluate corporate environmental performance and divides the total sample into high and low environmental performance groups for subgroup regression. The regression results are shown

**Table 6. The heterogeneity analysis.**

|  | (1) | (2) | (3) | (4) | (5) | (6) |
|---|---|---|---|---|---|---|
|  | stock | stock | stock | stock | stock | stock |
| dig | 0.0103 | 0.0441*** | 0.0212 | 0.0420*** | 0.0434*** | 0.0457*** |
|  | (0.0112) | (0.0118) | (0.0160) | (0.0123) | (0.0146) | (0.0113) |
| _cons | -4.5935*** | -4.0587*** | -4.2469*** | -4.3399*** | -4.5597*** | -4.4120*** |
|  | (0.3256) | (0.3186) | (0.4071) | (0.2871) | (0.3635) | (0.2660) |
| Control | YES | YES | YES | YES | YES | YES |
| Firm_FE | YES | YES | YES | YES | YES | YES |
| Year_FE | YES | YES | YES | YES | YES | YES |
| Obs | 13895 | 23250 | 10291 | 29758 | 17244 | 17020 |
| r2_a | 0.8774 | 0.7863 | 0.8509 | 0.7942 | 0.7729 | 0.8229 |

in columns (1) and (2) of Table 6, respectively representing the regression outcomes for high and low environmental performance samples. It is found that digital transformation has a more significant effect on enhancing supply chain efficiency for companies with low environmental performance. These companies, likely due to lower initial supply chain efficiency, see more pronounced improvements from digital investments. They may achieve more substantial process optimization, cost savings, and operational efficiency enhancements post-digital transformation. Conversely, companies with high environmental performance might already possess high supply chain efficiency, and therefore the marginal improvements from digital investments are smaller.

**The impact of firm size differences on outcomes.** Second, the study examines whether the results vary across different company sizes. The total sample is divided into large and small companies for regression. Specifically, a company is categorized as large if its total assets exceed the industry annual average, otherwise, it is considered small. The regression results are shown in columns (3) and (4) of Table 6, respectively for large and small companies. The effect of digital transformation on enhancing supply chain efficiency is more pronounced for smaller companies. This finding may indicate that smaller companies, with relatively limited resources, gain greater marginal benefits from digital transformation. These companies may not have fully realized process automation and informatization before transformation, hence the efficiency gains from digitalization are more evident. On the other hand, larger companies may have already achieved some degree of process optimization and automation, making the incremental benefits of digital transformation relatively limited.

**The impact of information disclosure quality differences on outcomes.** Lastly, the study examines whether the results change under different levels of information disclosure quality. The total sample is divided into high and low information disclosure quality groups for regression. Specifically, the study uses the KV information disclosure index to measure the quality of corporate information disclosure, with a lower KV index indicating higher quality. If the KV index is below the industry annual average, it is categorized as high information disclosure quality, otherwise, it is low. The regression results are shown in columns (5) and (6) of Table 6, respectively for high and low information disclosure quality companies. It is observed that digital transformation has a greater effect on enhancing supply chain efficiency for companies with low information disclosure quality. This suggests that the impact of digital transformation on improving supply chain efficiency is ubiquitous and not significantly affected by the level of corporate information disclosure. A possible explanation is that in companies with low information disclosure quality, there are issues of information asymmetry and opacity in internal management, which are alleviated through digital transformation. Digital

technologies such as ERP systems, cloud computing, and big data analytics can improve the availability and processing efficiency of information, reducing internal and external costs associated with information asymmetry. Therefore, these companies can improve supply chain efficiency more through digital transformation.

## Economic consequences analysis

The enhancement of supply chain efficiency is usually accompanied by the optimization of information processes, meaning that communication between businesses and suppliers, distributors, and customers becomes smoother. The result is a reduction in coordination costs due to information asymmetry or poor communication. Additionally, an efficient supply chain can provide more reliable delivery times and quality assurance, thereby reducing monitoring and enforcement costs during contract execution. Hence, improving supply chain efficiency helps to reduce a company's external transaction costs and enhance its market position and financial performance. To verify whether the increase in enterprise supply chain efficiency caused by digital transformation reduces the company's future external transaction costs, the study conducts the following two-stage economic consequences test, drawing on the approach by Kuang et al. (2023) [64]. As companies with high asset specificity face a higher risk of "lock-in" and are more likely to be exploited by trading partners, they face higher external transaction costs. Following the approach of Yuan et al. (2021) [65], the study uses the ratio of intangible assets to total assets to measure asset specificity (cost).

$$\Delta stock_{i,t} = \alpha_0 + \alpha_1 \Delta dig_{i,t} + \delta \Delta X + \gamma_i + \omega_t + \varepsilon_{i,t} \tag{3}$$

$$\Delta cost_{i,t+1} = \alpha_0 + \alpha_1 \Delta \hat{stock}_{i,t} + \delta \Delta X + \gamma_i + \omega_t + \varepsilon_{i,t} \tag{4}$$

The first and second-stage regression results are shown in columns (1) and (2) of Table 7, respectively. The coefficient in column (1) is significantly positive, indicating that a positive change in digital transformation leads to a positive change in enterprise supply chain efficiency. The coefficient in column (2) is significantly negative, indicating that a positive change in enterprise supply chain efficiency leads to a negative change in future external transaction costs. This result validates that the improvement in enterprise supply chain efficiency caused by digital transformation reduces the company's future external transaction costs. With the

**Table 7. The economic consequences analysis.**

|  | (1) | (2) |
|---|---|---|
|  | $\Delta stock_{i,t}$ | $\Delta cost_{i,t+1}$ |
| $\Delta dig_{i,t}$ | 0.0127*** |  |
|  | (0.0042) |  |
| $\Delta \hat{stock}_{i,t}$ |  | -0.0562*** |
|  |  | (0.0074) |
| _cons | 0.0210*** | 0.0156 |
|  | (0.0052) | (0.0105) |
| Control | YES | YES |
| Firm_FE | YES | YES |
| Year_FE | YES | YES |
| Obs | 34804 | 30744 |
| r2_a | 0.0063 | -0.0325 |

digitalization of the supply chain process, enterprises can more precisely track and forecast product flows, reducing additional costs caused by supply uncertainty.

## Discussion

This research extensively explores how digital transformation significantly enhances supply chain efficiency, especially within the context of global economic integration and intensifying market competition. We discovered that through the effective use of cutting-edge technologies such as the Internet of Things (IoT), cloud computing, big data analytics, and artificial intelligence, companies can not only optimize decision-making processes and resource allocation but also significantly increase the transparency of their supply chains. The application of these technologies, as demonstrated in the study by Paolucci et al. (2021) [66] in the automotive supply chain, not only improves cost efficiency but also optimizes the flow of information between businesses, suppliers, distributors, and customers, thereby boosting overall supply chain efficiency. Moreover, this study underscores the critical role of digital transformation in enhancing a company's rapid response capabilities to unforeseen events, such as a global pandemic, emphasizing the importance of immediate access and analysis of crucial supply chain data. This capability is vital for maintaining competitiveness in an ever-changing market environment.

Secondly, this study provides data-driven evidence of the two pathways through which digital transformation affects enterprise supply chain efficiency. Unlike existing literature that discusses the mechanisms of digital transformation on supply chain efficiency from perspectives of financing constraints, transaction costs, business efficiency, strategic layout [20, 21], this study finds that digital transformation can enhance supply chain efficiency by promoting internal corporate governance and by fostering external market competition. In terms of improving governance, digital transformation, by providing real-time data and analytical tools, strengthens monitoring processes and risk management [49], thereby improving supply chain transparency and collaborative efficiency [46]. In the aspect of external market competition, digital platforms and tools such as CRM and supply chain management software, enable businesses to capture market demand and consumer preferences more accurately, increasing market sensitivity and customer satisfaction, and thus enhancing enterprise supply chain efficiency [55]. This finding highlights the dual value of digital transformation in driving supply chain management, both improving internal management efficiency, and enhancing market responsiveness and customer satisfaction.

Our research further reveals that the impact of digital transformation on supply chain efficiency exhibits significant heterogeneity across different corporate backgrounds. Particularly, in companies with lower environmental performance, smaller scale, and lower quality of information disclosure, the contribution of digital transformation to supply chain efficiency is more pronounced. This indicates that digital transformation offers significant opportunities for efficiency improvements in companies with initially lower efficiency levels. Conversely, for those companies that have already achieved a higher level of supply chain management, the marginal benefits of digital transformation are relatively smaller. This finding underscores the necessity of refining digital investment strategies, especially for high-efficiency companies seeking to further optimize their supply chains through technological advancements. When implementing digital strategies, it is important to consider the specific needs and current efficiency levels of the company, ensuring that technology applications are targeted to address existing issues or enhance efficiency in key areas, thereby maximizing benefits. For example, for companies with high environmental performance, more precise and advanced digital technology applications, such as AI-driven demand forecasting and advanced data analytics, may be key to enhancing competitive advantages.

We further analyzed the economic consequences of digital transformation in enhancing supply chain efficiency, particularly in terms of reducing external transaction costs. By optimizing information flows, digital transformation enables more efficient and transparent communication between companies and their suppliers, distributors, and customers. This not only reduces coordination costs caused by information asymmetry or poor communication but also ensures the timing and quality of deliveries, significantly lowering the costs of monitoring and enforcement during contract execution. Therefore, enhancing supply chain efficiency is crucial for reducing a company's external transaction costs, thereby strengthening its market position and financial performance. To empirically test this, we employed a two-stage regression model to explore the relationship between digital transformation, supply chain efficiency enhancement, and a company's external transaction costs. The results show that digital transformation has a significant positive impact on corporate supply chain efficiency, which in turn significantly reduces a company's future external transaction costs. This finding highlights the importance of digital transformation in the current business environment, not only improving internal operational efficiency but also bringing significant economic benefits to the company.

The contributions of this article are mainly reflected in three aspects. First, in terms of research content, by directly exploring the impact of digital transformation on corporate supply chain efficiency rather than just traditional indicators like financial performance, this study provides new insights into the role of digital transformation in the field of supply chain management. Compared to existing literature, our research reveals a significant positive impact of digital transformation on supply chain efficiency, offering an important supplement to studies in this field and enriching the understanding of the comprehensive effects of digital transformation. Second, in terms of research mechanisms, by constructing a theoretical framework that includes internal governance levels and external market competition, this study reveals the specific mechanisms through which digital transformation promotes supply chain efficiency enhancement. This not only addresses the issue of unclear impact pathways in existing literature but also provides a new perspective for subsequent research analyzing the impact of digital transformation on corporate operations. Lastly, in terms of research expansion, this study explores the heterogeneous impact of digital transformation on supply chain efficiency across different corporate backgrounds. Through group regression analysis, we captured the variability of digital transformation effects, providing detailed guidance for different types of companies on how to achieve supply chain efficiency improvements during digital transformation. Furthermore, our analysis of economic consequences further examines how digital transformation reduces a company's future external transaction costs by enhancing supply chain efficiency, offering policy guidance for promoting digital transformation and supply chain management practices. Through these contributions, this study not only deepens the academic understanding of the impact of digital transformation but also provides practical insights and recommendations for corporate practice and policy formulation.

## Conclusion

### Research findings

With the acceleration of globalization and technological innovation, digital transformation has become key in driving enterprise competitiveness. Against this macro background, this study, based on data from A-share listed companies in China from 2007 to 2022, delves into how digital transformation impacts enterprise supply chain efficiency. The findings clearly indicate that digital transformation has a significant positive effect on enhancing supply chain efficiency, and robustness tests confirm the reliability of these results. Mechanism analysis reveals that the level of corporate governance and the degree of market competition are two primary

mediators through which digital transformation improves supply chain efficiency. Heterogeneity analysis shows that the impact of digital transformation on supply chain efficiency varies among companies with different environmental performances, sizes, and levels of information disclosure, with a more pronounced effect on companies with lower environmental performance, smaller size, and lower information disclosure quality. Lastly, the economic consequences analysis finds that digital transformation drives down future external transaction costs, thereby consolidating the enterprise's market competitiveness and elevating financial performance.

## Policy recommendations

Based on the findings of this study, we propose the following policy recommendations to promote digital transformation to further improve enterprise supply chain efficiency. First, given the significant positive impact of digital transformation on enterprise supply chain efficiency, enterprises and governments should cooperate to strengthen the construction of digital infrastructure, such as providing a stable network environment and efficient data management platform, and develop an integrated digital strategy that not only focuses on technology introduction itself, but also covers organizational culture, employee training and process optimization. To fully support efficient supply chain management. Second, given that improving internal governance is a key mechanism for improving supply chain efficiency through digital transformation, companies should aim to improve governance structures and ensure transparency and compliance in decision-making processes. The government should encourage enterprises to conduct technological research and innovation in key technology areas, such as artificial intelligence, big data analysis and the Internet of Things, by providing incentives such as policy and fiscal and tax incentives. In addition, policymakers should encourage collaboration and sharing of data and resources across the supply chain to improve efficiency, with a special focus on micro and small enterprises and smes, through tax breaks, financial support and professional training to help them invest in digital projects and promote the growth and development of these enterprises. Finally, while pursuing digital transformation, enterprises should consider their long-term development strategy, not only adapting to current market needs, but also anticipating future trends to ensure that the transformation strategy is sustainable and forward-looking. The government and industry organizations can provide information support to enterprises through the release of industry reports and trend forecasts to help them make more informed long-term development decisions. These policy recommendations aim to harness the potential of digital transformation more effectively and improve the efficiency of the corporate supply chain, thereby enhancing the market competitiveness and financial performance of enterprises, and are important for the continued health of enterprises themselves and the economy as a whole.

## Research limitations

While this study provides valuable insights into how digital transformation affects the efficiency of enterprise supply chains, we also recognize some limitations of the study. First of all, since the data is only from Chinese A-share listed companies, the results of the study may not be representative of all enterprises, especially small and medium-sized enterprises and non-listed companies. Second, this study does not fully consider the industry specificity, and may ignore the differences in digitalization needs and coping strategies of different industries. In addition, due to the focus on the Chinese market, the results of the study may not be applicable to other cultural and regional contexts. Rapid changes in technological development may also limit the timeliness of research conclusions. Future studies can make up for these limitations

by expanding the sample scope, introducing qualitative research methods, and designing causal experiments, so as to further enrich and deepen the understanding of the relationship between digital transformation and supply chain efficiency.

## Acknowledgments

The authors would like to thank the anonymous reviewers and the editors.

## Author Contributions

**Conceptualization:** Junbo He, Yaojun Fan.

**Data curation:** Junbo He, Min Fan.

**Formal analysis:** Junbo He, Yaojun Fan.

**Funding acquisition:** Junbo He, Yaojun Fan.

**Methodology:** Min Fan.

**Software:** Min Fan, Yaojun Fan.

**Supervision:** Yaojun Fan.

**Writing – original draft:** Junbo He, Min Fan, Yaojun Fan.

**Writing – review & editing:** Min Fan.

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
