## [Decision Letter · Decision Letter 0]

7 Feb 2024

PONE-D-23-42214Digital Transformation and Supply Chain Efficiency Improvement: An Empirical Study from A-Share Listed Companies in ChinaPLOS ONE

Dear Dr. Fan,

Thank you for submitting your manuscript to PLOS ONE. After careful consideration, we feel that it has merit but does not fully meet PLOS ONE’s publication criteria as it currently stands. Therefore, we invite you to submit a revised version of the manuscript that addresses the points raised during the review process.

After carefully reading the  current version of your manuscript, a few suggestions could be followed : please improve the link among the variables from the literature review, as to better support the hypotheses; within the Methodology and data analysis, please provide the thresholds as they were considered, with literature support; the same recommendation extends for the Results. Further, you need to clearly provide the status of your research Hypotheses and discuss it in the light of previous research. Limitations should also be considered for the current analysis. 

We look forward to receiving your revised manuscript.

Kind regards,

Ioana Gutu, Postdoctoral

Academic Editor

PLOS ONE

Journal Requirements:

Reviewers' comments:

Reviewer's Responses to Questions

**Comments to the Author**

1. Is the manuscript technically sound, and do the data support the conclusions?

Reviewer #1: Partly

Reviewer #2: Yes

Reviewer #3: Yes

2. Has the statistical analysis been performed appropriately and rigorously? 

Reviewer #1: N/A

Reviewer #2: No

Reviewer #3: Yes

3. Have the authors made all data underlying the findings in their manuscript fully available?

Reviewer #1: Yes

Reviewer #2: Yes

Reviewer #3: Yes

4. Is the manuscript presented in an intelligible fashion and written in standard English?

Reviewer #1: Yes

Reviewer #2: Yes

Reviewer #3: Yes

5. Review Comments to the Author

Reviewer #1: Comment 1: The introduction delves into generalities about the importance of an efficient supply chain without delving into the specific objectives or focus of the study until later. A more explicit statement about the research question, objectives, or hypotheses would make the introduction more focused and engaging.

Comment 2: While the introduction references existing literature on digital transformation, it falls short in explicitly identifying the gap that this study aims to fill. Clearly stating the research gap would strengthen the rationale for conducting the study and demonstrate the paper's contribution to the field.

Comment 3: While the review covers the basics of supply chain efficiency and the integration of advanced technologies, it lacks in-depth analysis and critical evaluation of existing literature. A more critical examination of methodologies, limitations, and gaps in the literature would strengthen the review.

Comment 4: The literature review identifies a need for enrichment and expansion in research perspectives, mechanisms, and scope regarding the relationship between digital transformation and supply chain efficiency. While this is valid, it would be helpful to explicitly state specific research questions or areas that future studies could explore.

the authors can rely on some other references to enrich there study in the literature and methodology:

- Khatib, S. F., Ismail, I. H., Salameh, N., Abbas, A. F., Bazhair, A. H., & Sulimany, H. G. H. (2023). Carbon emission and firm performance: The moderating role of management environmental training. Sustainability, 15(13), 10485.

- Ye, X., & Yue, P. (2023). Financial literacy and household energy efficiency: An analysis of credit market and supply chain. Finance Research Letters, 52, 103563.

Comment 5: The hypotheses (H1, H2, H3) are briefly stated but lack clarity in terms of how they will be tested or validated. Providing more explicit details on the variables involved in each hypothesis and the expected outcomes would enhance the rigor of the research framework.

Comment 6: The findings are reiterated in the introduction of the discussion section. Instead of restating the positive impact of digital transformation, consider delving directly into the interpretation and implications of the results to maintain reader engagement.

Comment 7: The study outlines its contributions in terms of content, mechanism, and expansion, but it would be beneficial to explicitly link each contribution to the specific findings or results discussed in the study. This would provide a clearer understanding of how each contribution is substantiated.

Comment 8: In the research findings section ensure that the policy recommendations directly align with the research findings. Explicitly state how each recommendation is derived from the study's results, reinforcing the practical implications of the research.

Comment 9: Integrate the limitations mentioned in the discussion section into the conclusion. Addressing limitations in the conclusion provides a holistic perspective on the study's scope and potential areas for improvement in future research.

Reviewer #2: ” Digital Transformation and Supply Chain Efficiency Improvement: An Empirical Study from A-Share Listed Companies in China” is an interesting paper, on a relevant topic, using a decent methodology. The information is easy to navigate, and the structure of the paper allows readers to analyse the concepts approached, providing an interesting insight of the topic. The paper is written according to academic standards, using proper language and scientific style.

However, before acceptance, the authors should pay attention to the following:

- The methodology lacks better presentation and detail. The reader is not informed what type of methodology is employed. A short mentioning is made only in section 5 Discussion:” Through group regression analysis, this paper captures the heterogeneity of the effects of digital transformation, providing detailed guidance for different types of enterprises on how to achieve supply chain efficiency improvement in the context of digital transformation. At the same time, through economic consequences analysis, the research also examines how digital transformation can reduce future external transaction costs by improving corporate supply chain efficiency, which has important policy guidance value for advancing digital transformation and supply chain management practices.”

Authors should develop this information in the Research methodology section and also briefly mention it in the Introduction part.

- For equations (1) and (2) the authors do not provide references. Moreover, at the beginning of section 3.3 Model Construction, authors state that:” Drawing on previous research...” and no reference is made to previous research that was analysed here.

Authors should develop the Research methodology part and include all necessary references that were investigated to build the model.

Reviewer #3: Dear Editor,

I reviewed the paper entitled Digital Transformation and Supply Chain Efficiency Improvement: An Empirical Study from A-Share Listed Companies in China and I have some things that concern me.

The paper has many strong points that, at first, convinced me it can be published after a minor revision.

These are:

1. The objective that can generate the interest of the readers and the arguments of the authors regarding the originality of the approach.

2. Literature review that is presented in an easy-to-follow way and that supports the research objective.

3. The methodology used, which considers several steps through which the authors solve specific problems such as: heterogeneity, endogeneity, etc.

4. Consistent database and varied variables, as well as the concern for checking the robustness of the results.

5. Correlation of own results with recent ones and confirmation of hypotheses.

6. Correlation of results with public policy proposals.

7. Accepting the existence of limitations.

However, checking if the authors have identified the most recent publications on the researched topic, I found a paper which does not appear in the references: Feimei Liao, Yaoyao Hu, Mengjie Chen, Shulin Xu, Digital transformation and corporate green supply chain efficiency: Evidence from China, Economic Analysis and Policy, Volume 81, 2024, Pages 195-207, ISSN 0313-5926, https://doi.org/10.1016/j.eap.2023.11.033.

The similarities between the reviewed paper and the one cited above are quite concerning, starting with the abstract. For this reason, I cannot decide on this article. I believe that the decision should be made by the editor according to the journal policy.

Thank you for understanding!

6. PLOS authors have the option to publish the peer review history of their article (what does this mean?). If published, this will include your full peer review and any attached files.

Reviewer #1: No

Reviewer #2: No

Reviewer #3: No

---

## [Author Response · Author response to Decision Letter 0]

27 Feb 2024

Dear Dr. Ioana Gutu and reviewers,

Firstly, I would like to express my sincere gratitude for your valuable comments during the review process. Your professional advice was crucial to our study, helping us identify the shortcomings in our paper and guiding us through a comprehensive revision. We have thoroughly revised our manuscript based on the comments from the reviewers and yourself, paying special attention to enhancing the connections between variables, clarifying our methodology, clearly stating our research hypotheses and discussing their relation to previous studies, as well as considering the limitations of our research in detail. Below are our responses to each reviewer's comments:

Response to Reviewer #1: Thank you for your meticulous review and valuable suggestions.

Regarding comment 1: Thank you for pointing out the issues in the introduction. We have revised the introduction, now clearly stating the research problem, objectives, and hypotheses, making it more focused and engaging.

Regarding comment 2: We appreciate your pointing out the lack of a clear research gap in the literature review. Following your suggestion, we have now clearly identified the specific research gap our study aims to fill, and strengthened the rationale for conducting our research.

Regarding comment 3: We acknowledge the previous lack of depth in the literature review. Now, we have added a critical evaluation of key literature, including discussions on limitations and future research directions.

Response to comment 4: Thank you for suggesting that the literature review could further enrich and expand on the research perspectives, mechanisms, and scope. Based on your advice, we have detailed specific issues and areas future research could explore, especially regarding the in-depth mechanisms between digital transformation and supply chain efficiency, and applications across different industry backgrounds.

Response to comment 5: Thank you for your suggestion on the clarity of hypothesis statements. We have revisited and clearly articulated each hypothesis's variables and expected outcomes, ensuring the research framework's logical coherence and the verifiability of the hypotheses.

Response to comment 6: We appreciate your suggestion about the repetition of results at the beginning of the discussion section. To avoid redundancy and maintain the reader's interest, we now directly proceed to interpret the results and their significance for existing research and practice, enhancing the value and depth of the discussion section.

Response to comment 7: We recognize the importance of clearly linking each research contribution to specific findings for the reader's understanding of the entire paper. Therefore, we have revised the conclusion section to ensure that each contribution is closely related to the research findings and clearly demonstrates how these contributions advance our understanding of the relationship between digital transformation and supply chain efficiency.

Response to comment 8: We are very grateful for your correction on the direct correspondence between policy suggestions and research findings. In the revised manuscript, we ensured that each policy suggestion is based on research findings and clearly indicated its source, strengthening the practical application value of our study.

Response to comment 9: We agree with the importance of discussing research limitations in the conclusion section. Therefore, we have comprehensively discussed the limitations of our study in the conclusion section, pointing out the potential impact of these limitations on the research conclusions, while also suggesting directions for future research to further explore this field.

Response to Reviewer #2: We are very grateful for Reviewer #2's specific suggestions on our methodology and introduction section. Regarding the enrichment of the introduction: We briefly mentioned the methodology adopted in our study in the introduction section.

Regarding the details of the methodology: We have expanded the methodology section, providing more details and references, and clarified the choice and implementation process of the research design.

Regarding the citation of equations: We have added specific literature citations for equations (1) and (2) and detailed the theoretical foundation of the model construction and references to previous research.

Response to Reviewer #3: We appreciate Reviewer #3's concerns and have conducted a thorough review regarding the similarity issue. Regarding the concern about similarity: We carefully compared the two papers and ensured our research provides unique insights and new evidence supplementing existing research. We also added the articles you mentioned in our references to show our attention to the latest research trends.

Overall, we have thoroughly revised our manuscript to ensure it meets the publication standards of PLOS ONE. We sincerely hope our responses and revisions address all concerns raised by you and the reviewers.

Once again, thank you for your valuable time and professional advice.

Sincerely, Min Fan

---

## [Decision Letter · Decision Letter 1]

19 Mar 2024

PONE-D-23-42214R1Digital Transformation and Supply Chain Efficiency Improvement: An Empirical Study from A-Share Listed Companies in ChinaPLOS ONE

Dear Dr. Fan,

Thank you for submitting your manuscript to PLOS ONE. After careful consideration, we feel that it has merit but does not fully meet PLOS ONE’s publication criteria as it currently stands. Therefore, we invite you to submit a revised version of the manuscript that addresses the points raised during the review process.

**Please consider clearly and concisely addressing all the concepts and variables that define your research, along with other minor issues as kindly recommended by Reviewers.**

We look forward to receiving your revised manuscript.

Kind regards,

Ioana Gutu, Postdoctoral

Academic Editor

PLOS ONE

Journal Requirements:

Reviewers' comments:

Reviewer's Responses to Questions

**Comments to the Author**

1. If the authors have adequately addressed your comments raised in a previous round of review and you feel that this manuscript is now acceptable for publication, you may indicate that here to bypass the “Comments to the Author” section, enter your conflict of interest statement in the “Confidential to Editor” section, and submit your "Accept" recommendation.

Reviewer #1: All comments have been addressed

Reviewer #4: All comments have been addressed

2. Is the manuscript technically sound, and do the data support the conclusions?

Reviewer #1: Yes

Reviewer #4: Yes

3. Has the statistical analysis been performed appropriately and rigorously? 

Reviewer #1: Yes

Reviewer #4: Yes

4. Have the authors made all data underlying the findings in their manuscript fully available?

Reviewer #1: Yes

Reviewer #4: Yes

5. Is the manuscript presented in an intelligible fashion and written in standard English?

Reviewer #1: Yes

Reviewer #4: No

6. Review Comments to the Author

Reviewer #1: I strongly recommend that we submit this paper for publication. The depth of research, clarity of arguments, and significance of findings make it an ideal candidate for dissemination within the academic community. The meticulous attention to detail, thorough analysis, and innovative approach showcased throughout the manuscript demonstrate the dedication and expertise of the authors. By sharing this work with the broader scientific community, we not only contribute to the advancement of knowledge in our field but also invite valuable feedback and discussion that can further refine and strengthen our conclusions. Therefore, I believe it is imperative that we seize this opportunity to disseminate our findings through publication, ultimately enriching the scholarly discourse and advancing our collective understanding

Reviewer #4: Based on the paper's content, the paper is well executed but would benefit from several improvements.

1 Regarding "internal corporate governance" and "external market competition" as two major mechanisms of digital transformation, ensure that the most recent and relevant literature is cited to reinforce the authority of these concepts.

2 In the abstract, the statement that "digital transformation significantly enhances supply chain efficiency" should be revised to "the research indicates that digital transformation plays a key role in significantly enhancing supply chain efficiency" to increase the accuracy of the semantics.

3 Review all cited references in the paper, especially in the abstract and conclusion sections, to ensure consistency and accuracy of the citation format, such as author names, publication years, and paper titles following prescribed guidelines.

4 When discussing how digital transformation impacts supply chain efficiency across different corporate backgrounds, consider optimizing the structure of this section to make it more logical and organized. Clear subheadings could be used to distinguish between different aspects, providing better clarity.

5 Carefully check the citation format and ensure uniformity and standardization of all figures and image annotations throughout the document.

6 For the conclusion section that states "the improvement in supply chain efficiency caused by digital transformation can reduce future external transaction costs and enhance the market position and financial performance of the enterprise," it is recommended to rephrase more concisely as "digital transformation drives down future external transaction costs, thereby consolidating the enterprise's market competitiveness and elevating financial performance," to emphasize its direct positive impact on the company's performance.

I hope it helps with your research. Good luck.

7. PLOS authors have the option to publish the peer review history of their article (what does this mean?). If published, this will include your full peer review and any attached files.

Reviewer #1: No

Reviewer #4: No

---

## [Author Response · Author response to Decision Letter 1]

22 Mar 2024

Dear Dr. Ioana Gutu and Reviewers,

We are profoundly grateful for the opportunity to revise our manuscript and resubmit it for consideration for publication in PLOS ONE. We deeply appreciate the constructive feedback provided by the academic editor and reviewers. Their insights have been invaluable in enhancing the quality and clarity of our work. We have conducted a comprehensive review and a more detailed description of the main concepts and variables used in our research, ensuring the clarity and logic of our research framework and results.

Reviewer #1:

First and foremost, we wish to express our sincerest gratitude to Reviewer #1 for their positive evaluation of our manuscript and for recognizing the depth and significance of our research.

Reviewer #4:

We also thank Reviewer #4 for their detailed and constructive feedback, which has played a significant role in improving our manuscript. Here are our responses to each of the points raised:

Literature on Digital Transformation Mechanisms: We have added two more recent and relevant literature references to strengthen the authority of these concepts. Thank you for pointing this out, which has made our theoretical framework more robust.

Abstract Semantics: The statement in the abstract has been revised to "the research indicates that digital transformation plays a key role in significantly enhancing supply chain efficiency," as suggested. This alteration better captures the essence of our findings and aligns with the empirical evidence presented.

Consistency in Citation Format: Following your suggestion, we have carefully reviewed and corrected the citation format throughout the manuscript, ensuring the consistency and accuracy of all references according to the prescribed guidelines. This includes a meticulous check of author names, publication years, and paper titles.

Organizational Structure: We have reorganized the section discussing the impact of digital transformation across different corporate backgrounds, implementing clear subheadings to differentiate various aspects. This structure provides better clarity and logic, enhancing the reading experience.

Standardization of Figures and Images: All figures and image annotations have been carefully checked and standardized to ensure the uniformity of the entire document. This standardization enhances the visual presentation and coherence of our research findings.

Concise Conclusion: The conclusion section has been revised to more concisely state the direct positive impact of digital transformation on company performance. The new phrasing, "digital transformation drives down future external transaction costs, thereby consolidating the enterprise's market competitiveness and elevating financial performance," succinctly encapsulates our core message and findings.

We hope our revisions and responses adequately address the concerns and comments raised by the reviewers.

---

## [Editor Report · Decision Letter 2]

28 Mar 2024

Digital Transformation and Supply Chain Efficiency Improvement: An Empirical Study from A-Share Listed Companies in China

PONE-D-23-42214R2

Dear Dr. Min Fan,,

We’re pleased to inform you that your manuscript has been judged scientifically suitable for publication and will be formally accepted for publication once it meets all outstanding technical requirements.

Kind regards,

Ioana Gutu, Postdoctoral

Academic Editor

PLOS ONE
---

## [Editor Report · Acceptance letter]

3 Apr 2024

PONE-D-23-42214R2 

PLOS ONE

Dear Dr. Fan, 

I'm pleased to inform you that your manuscript has been deemed suitable for publication in PLOS ONE. Congratulations! Your manuscript is now being handed over to our production team.

Kind regards, 

on behalf of

Dr. Ioana Gutu 

Academic Editor

PLOS ONE